Long non-coding RNA NMRAL2P promotes glycolysis and reduces ROS in head and neck tumors by interacting with the ENO1 protein and promoting GPX2 transcription

Nie Qian
Cao Huan
Yang JianWang
Liu Tao
Wang BaoShan hebwangbs@163.com
Department of Otorhinolaryngology, Second Hospital of Hebei Medical University , Shijiazhuang , China
Orlov Yuriy
Electronic publication date: 2023 Oct 2
Publication date: 2023
Volume: 11
Electronic Location ID: e16140
Received 2023 Apr 25; Accepted 2023 Aug 29
Copyright: ©2023 Nie et al.
Copyright year: 2023
Copyright holder: Nie et al.
License: This is an open access article distributed under the terms of the Creative Commons Attribution License, which permits unrestricted use, distribution, reproduction and adaptation in any medium and for any purpose provided that it is properly attributed. For attribution, the original author(s), title, publication source (PeerJ) and either DOI or URL of the article must be cited.
License URL: https://creativecommons.org/licenses/by/4.0/

Keywords: NMRAL2P, GPX2, ENO1, Glycolysis, ROS

Funding: National Natural Science Foundation of China 81972553 This work was supported by the Project of the National Natural Science Foundation of China (Study on the mechanism of epigenetic regulation of SOCS1 and ATF3 expression by DNA hypomethylation activated LINC00668 in promoter region to promote invasion and metastasis of laryngeal squamous cell carcinoma: 81972553). The funders had no role in study design, data collection and analysis, decision to publish, or preparation of the manuscript.

==============================
Background

Metabolic reprogramming is a key marker in the occurrence and development of tumors. This process generates more reactive oxygen species (ROS), promoting the development of oxidative stress. To prevent ROS from harming tumor cells, tumor cells can increase the production of reducing agents to counteract excessive ROS. NMRAL2P has been shown to promote the production of reductive mRNA and plays an important role in the process of oxidative stress.

Methods

In this study, the clinical data and RNA sequencing of head and neck tumors were obtained from The Cancer Genome Atlas data set. The long non-coding RNA (LncRNA) related to oxidative stress were then identified using differential and correlation analyses. The differential expression and prognosis of the identified lncRNA were then verified using samples from the library of the Second Hospital of Hebei Medical University. Only NMRAL2P was substantially expressed in cancer tissues and predicted a poor prognosis. The tumor-promoting impact of NMRAL2P was then confirmed using in vitro functional assays. The data set was then split into high- and low-expression subgroups based on the median gene expression of NMRAL2P to obtain the mRNA that had a large difference between the two groups, and examine the mechanism of NMRAL2P on GPX2 using quantitative real-time PCR, RNA binding protein immunoprecipitation assay, and chromatin immunoprecipitation. Mass spectrometry was used to identify NMRAL2P-binding proteins and western blotting was used to investigate probable mechanisms.

Results

The lncRNA NMRAL2P is associated with oxidative stress in head and neck tumors. In vitro functional assays showed that the gene has a cancer-promoting effect, increasing lactic acid and superoxide dismutase production, and reducing the production of ROS and malondialdehyde. NMRAL2P promotes the transcription of GPX2 by binding to transcription factor Nrf2. The gene also inhibits the degradation of ENO1, a crucial enzyme in glycolysis, by binding to protein ENO1.

Conclusions

This study shows that NMRAL2P can promote glycolysis and reduce the harm to tumor cells caused by ROS. The gene can also be used as a possible target for the treatment of head and neck tumors.

Introduction

The rapid growth of tumors requires considerable energy, and the way the mitochondrial tricarboxylic acid cycle produces ATP cannot keep up with these needs, so glycolysis is the primary way tumors obtain energy (Hsu & Sabatini, 2008; De Berardinis et al., 2008). Previous studies have also demonstrated that glycolysis is not a unique instance of hypoxia in tumor energy metabolism, but rather that tumors prioritize glycolysis even when there is enough oxygen to create energy for themselves (Pascale et al., 2020). Lactic acid is produced in significant quantities during glycolysis, and the acidic environment makes it easier for tumor cell immune escape (Chen et al., 2022). Tumor samples show positive changes to glycolysis-related enzymes. For example, the protein levels of GLUT-1, G6PI, LDHA, LDHB, and PFK-L increase, cervical tumor migration increases and the invasion ability of cancer is also enhanced (Chen et al., 2021). The RNA expression ratio of PKM2/PKM1 increases, which is an important pathway for colorectal cancer development (Lan et al., 2020). Another study showed that the protein expression of GLUT1 in prostate cancer tissue is higher than in adjacent tissues (Xiao et al., 2018). Metabolic reprogramming is also a double-edged sword, since glycolysis produces a large number of ROS, which, in the presence of an intracellular oxidant/reductant concentration imbalance, results in the activation of the oxidative stress process, culminating in stress-related DNA and protein damage, and ultimately cell death (Matsui et al., 2000; Park & Chung, 2019). By controlling the oxidation/reduction balance, tumor cells can proliferate quickly. In order to target tumors and treat cancer, identifying the adjustment point of the tumor is crucial.

Mechanisms such as metabolic reprogramming and oxidative stress trigger abnormalities in cancer tissues that impact numerous lncRNA (Tan et al., 2021; Ginckels & Holvoet, 2022; Zhang, Sun & Zhang, 2022). LncRNA perform a variety of tasks, including altering mRNA transcription by acting on the promoter region of the mRNA. LncRNA can also raise mRNA levels, which are the target of miRNA, by competitive binding with miRNA, and they can serve as a framework for constructed proteins or RNA (Luo et al., 2020; Tian et al., 2020; Lin et al., 2020). LncRNA are promising targets for tumor therapy, but there are currently not enough studies on lncRNA connected to oxidative stress in head and neck tumors.

The differentially expressed lncRNA (DE lncRNA) related to oxidative stress were identified in this study through differential and correlation analyses of the sequencing data of head and neck tumors in The Cancer Genome Atlas data set (TCGA). The DE lncRNA identified were then confirmed using 45 pairs of paired head and neck tumors from the library of the Second Hospital of Hebei Medical University. To further investigate the functional effect of related genes on the migration and invasion of head and neck tumors, human derived head and neck tumor TU177 and human derived head and neck metastatic lesion AMC-HN-8 were selected as cell lines for in vitro assays. The influence of lncRNA on carcinogenesis and development was further confirmed by analyzing cell functions and downstream mechanisms.

Materials and Methods

Data acquisition of bulk-sequencing and clinical data

A total of 546 RNA bulk sequencing and clinical data of head and neck tumors (tumor: 502; normal: 44) were downloaded from the GDC website (https/portal.gdc.cancer.gov), and 45 cancer samples and their matching samples (non-tumor tissue adjacent to the cancer) were obtained by the second hospital of Hebei Medical University. Before surgery, the patients did not receive chemotherapy or radiotherapy (clinical information is shown in Table 1). All the samples used in this study are from the biological sample database of the second Hospital of Hebei Medical University. The Research Ethics Committee of Hebei Medical University’s Second Hospital approved this study (2021-R012-01). Written, informed consent was obtained from all patients included in this study.

Table 1 Basic clinical information of patients.

Clinical data (N = 45)	
age	62.31 ± 7.80	
male	45 (100)	
stage		
I	10 (22.22)	
II	7 (15.56%)	
III	13 (28.89%)	
IV	15 (33.33%)	
T		
T1	12 (26.67%)	
T2	13 (28.89%)	
T3	12 (26.67%)	
T4	8 (17.78%)	
N		
N0	28 (62.22%)	
N1	8 (17.78%)	
N2	9 (20%)	
M		
M0	44 (97.78%)	
Mx	1 (2.22%)	

Cell culture and treatment

Head and neck tumor cell lines (TU177, AMC-HN-8) were purchased from Beijing’s Beina Chuanglian Institute of Biotechnology. The cell lines were cultured in RPM1640 or DMEM media with 10% fetal bovine serum at 37 °C and 5% CO2. All of the culture medium used in this study were obtained from GIBCO 108 (Gibco, Billings, MT, USA).

Plasmid acquisition, synthesis of antisense oligodeoxynucleotides (ASO)

NMRAL2P overexpression plasmid (NMRAL2P-oe) and pcDNA 3.1 empty plasmid (vector), which was used as the negative control, were both purchased from Sangon Biotech (Shanghai, China, Sangon). The verification results of NMRAL2P overexpression plasmid are presented in File S1. NMRAL2P-ASO (Antisense oligonucleotide; Ribobio, Guangzhou, China) was designed to knock down the expression of NMRAL2P, and NC-ASO, also provided by RIBOBIO biological company, was used as the negative control. ENO1 and GPX2 were knocked down by cloning the designed target sequence into the vector plasmid pGenesil-1. File S2 presents the sequence of NMRAL2P-ASO, shENO1, and shGPX2 and the sequencing results of shENO1 and shGPX2. ENO1 full-length plasmid (ENO1-FL) and truncated plasmid (ENO1-A, ENO1-B, ENO1-C) with flag tag sequence were purchased from GENEWIZ (Suzhou, China) and the sequencing certificate they provided can be found in File S3.

Total RNA extraction, reverse transcription, quantitative real-time polymerase chain reaction (QRT-PCR)

Total RNA was extracted from the tissues and cells using the EaStep® Super Total RNA extraction kit (LS1040; Promega, Madison, WI, USA). RNA concentration was measured and the quality was assessed using a spectrophotometer (ND-2000; Thermo Fisher, Waltham, MA, USA). The extracted total RNA was reverse transcribed using Roche’s Transcriptor First Strand cDNA Synthesis Kit to obtain the cDNA of a tissue or cell, and the GoTaq®qPCR Master Mix (A6001; Promega, Madison, WI, USA) kit was used for QRT-PCR. The data were normalized using the 2-ΔΔCt technique with 18S serving as the internal reference. Table S1 shows the primer sequences used in QRT-PCR.

Nuclear-cytoplasmic separation

Nuclear and cytoplasmic components were separated, and the total RNA of the nucleus and cytoplasm were obtained by NE-PER isolation (78833, Thermo Fisher, California, USA), and then reversed to produce cDNA. The subcellular localization of NMRAL2P was examined with QRT-PCR assays using U6 and GAPDH as references.

Cell transfection

TU177 and AMC-HN-8 were cultured in a six-well plate. When the fusion rate reached 80%–90%, 4 µg NMRAL2P-oe/ 5 µg NMRAL2P-ASO and its corresponding negative control (pcDNA3.1, ASO-NC) were combined with 7.5 µg Hieff Trans Liposomal Transfection Reagent (40802ES03; Yeasen Biotechnology, Shanghai, China), and then transferred into TU177 and AMC-HN-8 cells. The transfection efficiency was observed after 48 h, and the cells were collected for further assays.

Cell proliferation assays

After 48 h of transfection, TU177 and AMC-HN-8 cells were digested into cell suspension by trypsin, and then 2*103 (2,000) cells were inoculated into a 96-well plate. To reduce random errors, all experiments were performed in triplicate. After 0 h, 24 h, 48 h and 72 h, 20 µl MTS (G3580; Promega) was introduced into the experimental wells. After being cultured in the incubator for 3 h, Spark®143 multimode microplate reader (Tecan, Männedorf, Switzerland) was used to detect the absorbance of the samples in 490 nm.

Invasion and migration assays

Transwell migration and invasion assays were used to observe the migration and invasion of tumor cells (CLS3396; Corning, New York, USA). Matrigel (CLS354483; Corning) was removed from the −80 °C refrigerator, placed on an ice plate, and then diluted with a serum-free medium at a 1:14 ratio. A precooling gun was used to add 50 µL diluted Matrigel (about 27 µg) to the upper chamber of transwell before placing it in a 37 °C incubator for more than 30 min to ensure that the matrix adhesive was fully solidified. A total of 650 µL of medium containing 10% fetal bovine serum was added to the lower chamber in the migration and invasion assays. Following 48 h of transfection, TU177 and AMC-HN-8 were digested with trypsin, and then 100 µL of serum-free medium, supplemented with 1*105 (100,000) transfected cells, was introduced into the upper chamber. After culturing for 24 h in a 37 °C incubator, the non-invading or non-migrating cells in the upper chamber were carefully removed, and then the chamber was placed in 4% paraformaldehyde and fixed for 30 min, washed with PBS three times, and stained with crystal violet for 30 min. After the crystal violet was removed, washed, and dried, the cells were imaged and then counted in at least three random fields under a microscope.

Cell colony formation assay

TU177 and AMC-HN-8 cells were digested into cell suspension by trypsin 48 h after the migration and invasion assays were performed and 2*103 cells were then planted in a six-well plate. The culture medium was removed from a frosix-well plate after 14 days, the PBS was washed three times, and 4% paraformaldehyde was fixed for 20 min, followed by 20 min of 0.5% crystal violet staining. The number of clones was counted after the crystal violet was removed, washed, and dried. All assays were performed in triplicate.

Measurement of ROS production and lactic acid production

The Lactic Acid Content Assay Kit from Solarbio (Micro method, BC2235; Solarbio, Beijing, China) was used to measure the concentration of lactic acid. The transfected cells were collected, the extract was added, and the supernatant was collected. The cells were then crushed in an ice bath (power: 300W, ultrasonic: 3 s, interval: 7 s, total time: 3 min), and the supernatant was collected after centrifugation. The experimental reagent was then added according to the manufacturer’s instructions. The sample’s absorbance at 570 nm was detected using Spark® multimode microplate reader, and the concentration of lactic acid was calculated. The Reactive Oxygen Species Assay Kit (CA1410; Solarbio, Beijing, China) was used to measure ROS content. Following transfection, the cell culture medium was taken out and 1 ml of diluted DCFH-DA (2,7-dichlorofluorescein diacetate; reactive oxygen species fluorescent probe) was added. For 20 min, the cells were cultured in a cell incubator at 37 °C. To completely eliminate the DCFH-DA that did not enter the cells, the cells were washed with serum-free cell culture medium three times. The fluorescence intensity before and after stimulation was measured using the excitation wavelength of 488 nm to determine the ROS content.

Measurement of malondialdehyde content and superoxide dismutase activity

The Malondialdehyde (MDA) Content kit (Micromethod, BC0025; Solarbio, Beijing, China) was used to measure MDA content. The transfected cells were collected, the extract was added, and the supernatant was collected. The transfected cells were crushed in an ice bath (power: 200 W, ultrasound: 3 s, interval: 10 s, repeated: 30 times), and the supernatant was collected after centrifugation. The experimental reagent was then added according to the manufacturer’s instructions. The absorbance of the sample at 534 nm and 600 nm was then measured using the Spark®143 multimode microplate reader, and the MDA content was calculated. The Superoxide Dismutase Activity Detection Kit (Micro method, BC0175; Solarbio, Beijing, China) was used to detect SOD activity. The supernatant of broken cells was collected and the experimental reagent was added according to the manufacturer’s instructions. Finally, the absorbance of the sample at 560 nm was detected using the Spark®143 multimode microplate reader, and the SOD activity was calculated.

RNA binding protein immunoprecipitation assay (RIP)

The PureBinding®RNA Immunoprecipitation Kit was used to perform RIP assays (P0102; Geneseed, Guangzhou, China). TU177 or AMC-HN-8 cells were seeded in a 10-cm dish at 70%–80% confluence, harvested by trypsinization, and lysed with RIP cell lysate. The cell lysates were added to the target protein antibody or the magnetic bead coupling complex of Anti-Rabbit IgG, respectively, and incubated overnight at 4 °C. The precipitated RNA from each group was extracted via washing, adsorption, and purification, and the cDNA was subsequently obtained using reverse transcription. The relevant verification experiments were then carried out using QRT-PCR. All the experiments were repeated, independently, three times.

Chromatin Immunoprecipitation Assay (ChIP)

The EZ-Magna ChIP®A/G Chromatin Immunoprecipitation Kit (17-10086; Millipore, Burlington, MA, USA) was used to perform ChIP assays. TU177 cells were seeded in a 10-cm dish, then transferred into NMRAL2P-oe and the vector. After 48 h, the cells were collected with trypsin, and then formaldehyde was cross-linked for 10 min. Next, the lysate and ultrasonic treatment were added to an average size of 300–500 base pairs and then fixed with Nrf2 antibody (80593-1-RR; Proteintech, Wuhan, China) magnetic beads complex to form a protein-DNA complex. The DNA from the sample was then washed out of the complex, and then the DNA was verified through ChIP-QPCR. All the experiments were repeated, independently, three times.

MS2-12X-dependent RNA pull-down assay

We used BamHI and EcoR I to digest MS2-12X and NMRAL2P-oe. Gel electrophoresis was used to check whether the digestion was successful, and the plasmids and fragments were recovered. The 12X fragment was ligated with NMEAL2P-oe with T 4 ligase, and sequencing was used to check whether the connection is successful. The corresponding sequencing results are shown in File S2. Finally, the NMRAL2P-oe plasmid and PSL-MS2 plasmid with 12X sequence were simultaneously transferred into TU177 cells at a 1:1 ratio, and then the proteins of the GFP antibody group and negative control lgG antibody group were collected with a RIP assay and then analyzed using mass spectrometry.

Mass spectrometry was performed using the methods described by Chen et al. (2019): the protein used for mass spectrometry was hydrolyzed by FASP, and then the peptide mixture was loaded on a reverse phase trap column (Thermo Fisher Scientific Acclaim PepMap100, 100 µm*2 cm, nanoViper C18) and connected to a C18-reversed phase analytical column (Thermo Fisher Scientific Easy Column, 10 cm long, 75 µm inner diameter, 3 µm resin; Thermo Fisher Scientific, Waltham, MA, USA) in buffer A (0.1% formic acid) and separated with a linear gradient of buffer B (84% acetonitrile and 0.1% formic acid; flow rate was 300 nL/min). After separation, the samples were analyzed by LC-MS/MS on a Q Exactive mass spectrometer (Thermo Fisher Scientific). The LC-MS/MS spectrum and MS/MS spectra were analyzed using the MASCOT engine (Matrix Science, UK, Version 2.2). The following settings were used for protein identification: peptide mass tolerance = 20 ppm, MS/MS tolerance = 0.1 Da, enzyme = trypsin, missed cleavage = 2, fixed modification: Carbamidomethyl (C), and variable modification: oxidation (M).

Western blot

Following 48 h of transfection, 200 µL of lysate was added to the six-well plate (RIPA lysate: R0010; Solarbio, Beijing, China; PMSF: P0100; Solarbio, Beijing, China; protease inhibitor: G6521, Promega, Madison, WI, USA; using a 100:1:1 ratio). The cell protein lysate was mixed with protein sample loading buffer (LT103; Epizyme, Shanghai, China) at a 4:1 ratio, and then boiled for 5 min at 95 °C in a metal bath. An identical volume of (25 µg) protein was added to the 10% sodium dodecyl sulfate-polyacrylamide gel electrophoresis plate (SDS-PAGE) for electrophoresis. The protein was then transferred to the polyvinylidene fluoride membrane. The membrane was sealed with 5% bovine serum albumin for 2 h and then incubated with corresponding primary antibodies at 4 °C overnight. The sample was rinsed with TBST the next day and then incubated with the corresponding secondary antibody at ambient temperature for 2 h. Chemiluminescence was displayed using Fluorescent XRS+ (Bio-Rad, Hercules, CA, USA). β-actin served as the internal reference, the first antibody was anti-ENO1 (1:1000, ET1705-56; Huabio, Hangzhou, China) and anti- β actin (1:10000; 81115-1-RR; Proteintech, Wuhan, China). The second antibody was Anti-Rabbit IgG (Haul) (1:10000, SA00001-2; Proteintech, Wuhan, China)

Statistical analysis

SPSS (version 16.0; Chicago, IL, USA) and R were used for the statistical analysis (version 4.1.2; R Core Team, 2021). Graphs were generated with GraphPad Prism software (version 7.0; GraphPad Inc., San Diego, CA, USA). T-test was used to assess the differences between groups of numerical variables with normal distribution, and the rank sum test was employed if the numerical variable deviated from the normal distribution. The proliferation curve was statistically analyzed by two-way ANOVA. The classified variables were the chi-square test or Fisher exact probability method, and a Kaplan–Meier analysis was used for a survival analysis.

Result

Identifying the lncRNA involved in regulating oxidative stress

RNA-seq and clinical data of 546 head and neck tumors were obtained from TCGA, including 502 tumor samples and 44 normal samples. The “limma” package was used to analyze the differentially expressed mRNA (DE mRNA) and DE lncRNA between the normal samples and cancer samples. A total of 1,421 DE mRNA were obtained through the differential analysis, of which 537 were upregulated and 884 were downregulated in head and neck tumors (Fig. 1A). A total of 451 DE lncRNA were obtained, including 252 upregulated lncRNA and 199 downregulated lncRNA (Fig. 1B). A total of 1,399 mRNA associated with oxidative stress were retrieved from Genecards (Table S2), of which 113 were differentially expressed in head and neck tumor samples (Fig. 1C).

Figure 1 Screening of lncRNA related to oxidative stress.

(A) Differential analysis of mRNA expression in head and neck tumor samples. (B) Differential analysis of lncRNA expression in head and neck tumor samples. (C) Oxidative stress-related mRNA. (D) Oxidative stress-related lncRNA. (E) Differential expression of NMRAL2P in 45 pairs of clinical samples. (F) Prognostic analysis of NMRAL2P in 45 pairs of clinical samples.

A Pearson correlation analysis was then performed between the 113 DE mRNA related to oxidative stress and 451 DE lncRNA, and 16 DE lncRNA associated with oxidative stress were screened (correlation coefficient > 0.7 and P Value < 0.001; Fig. 1D).

NMRAL2P was differentially expressed in 45 pairs of samples and was associated with poor prognosis

A total of 45 pairs of head and neck samples from the Second Hospital of Hebei Medical University were used as the verification set to confirm the differential expression of the 16 identified DE lncRNA to screen those associated with oxidative stress. Only NMRAL2P was highly expressed in head and neck tumors (Fig. 1E) and high expression of the gene was associated with poor prognosis (Fig. 1F). Subcellular localization showed that NMRAL2P is mainly located in the nucleus, accounting for approximately 60% of the total NMRAL2P found in the cell (Fig. 2A).

Figure 2 Functional assays of NMRAL2P overexpression.

(A) Subcellular localization of NMRAL2P in cancer cells. (B) Verification of NMRAL2P-oe overexpression efficiency in TU177 and AMC-HN-8 cells by QRT-PCR. (C–D) Changes in TU177 and AMC-HN-8 cell viability after being transfected with NMRAL2P-oe, verified by MTS assay. (E–F) Changes in cell migration and invasion ability of TU177 and AMC-HN-8 after being transfected with NMRAL2P-oe, verified by Transwell assay. (G–H) Changes in proliferation activity of TU177 and AMC-HN-8 after being transfected with NMRAL2P-oe, verified by clone formation assay. (I–J) Changes in lactic acid level, ROS yield, SOD activity and MAD content of TU177 and AMC-HN-8 after being transfected with NMRAL2P- oe. (ROS, Reactive Oxygen Species; SOD, Superoxide Dismutase; MAD, Malondialdehyde; NMRAL2P-oe, Overexpression of NMRAL2P Vector:pcDNA3.1).

NMRAL2P affects the migration, invasion, and proliferation of head and neck tumor cells in vitro

NMRAL2P-oe was transferred into TU177 and AMC-HN-8 cells, respectively, to further study the impact of NMRAL2P on the function of head and neck tumor cells. The overexpression of NMRAL2P-oe was verified using QRT-PCR. Compared with the control group, the expression of NMRAL2P increased around 100x in the overexpression group (Fig. 2B), and compared to the vector, the NMRAL2P overexpression increased the viability, migration, invasion, and proliferation of tumor cells (Figs. 2C–2H). To reverse-verify the function of NMRAL2P in head and neck tumor cells, we transferred NMRAL2P-ASO into TU177 and AMC-HN-8 cells, respectively, and the relative expression of NMRAL2P in TU177 and AMC-HN-8 was knocked down by more than 40% (Figs. 3A, 3B). The knockout efficiency of NMRAL2P-ASO met the requirements for functional assays. During the assays, it was observed that the survival, migration, invasion, and proliferation of cells in the knockdown group were substantially poorer than that of the negative control group (NC; Figs. 3C–3H), verifying that the expression of NMRAL2P improves the viability, migration, invasion, and proliferation of head and neck tumor cells.

Figure 3 The functional assay of NMRAL2P knockdown.

(A–B) Verification of NMRAL2P-ASO knockdown efficiency in TU177 and AMC-HN-8 cells by QRT-PCR. (C–D) Changes in cell viability of TU177 and AMC-HN-8 after being transfected with NMRAL2P-ASO, verified by MTS assay. (E–F) Changes in cell migration and invasion ability of TU177 and AMC-HN-8 after being transfected with NMRAL2P- ASO, verified by Transwell assay. (G–H) Changes in proliferation activity of TU177 and AMC-HN-8 after being transfected with NMRAL2P-ASO, verified by clone formation assay. (I–J) Changes in lactic acid level, ROS yield, SOD activity and MAD content of TU177 and AMC-HN-8 after being transfected with NMRAL2P- ASO. (ASO, Antisense oligonucleotide; NC, NC-ASO, Negative control of NMRAL2P knockdown; NMRAL2P-ASO, Knocks down the expression of NMRAL2P; QRT-PCR, quantitative real time polymerase chain reaction).

NMRAL2P affects the production of lactic acid, ROS, SOD, and MDA

To further investigate whether NMRAL2P influences oxidative stress and glycolysis-related processes, we transferred NMRAL2P-oe and NMRAL2P-ASO into TU177 and AMC-HN-8 cells using the same experimental methods used previously, and measured lactic acid, SOD, MDA, and ROS using a micromethod and fluorescence probe. Increased expression of NMRAL2P led to an increase in lactic acid and SOD. Conversely, ROS and MDA in cells were negatively correlated with NMRAL2P expression, meaning ROS and MDA decreased as NMRAL2P expression increased (Figs. 2I, 2J). When NMRAL2P was knocked down, the opposite results were observed (Figs. 3I, 3J). These results indicate that NMRAL2P may promote the production of lactic acid and SOD and inhibit the production of ROS and MDA, suggesting that NMRAL2P can enhance glycolysis and reduce oxidative stress.

NMRAL2P binds to ENO1 and enhances its stability

In order to further investigate the mechanism of NMRAL2P, the 12X sequence was introduced into the GFP vector together with the NMRAL2P sequence (Fig. 4A). The protein was then brought down using the GFP vector and the lgG antibody for the mass spectrometry (MS) analysis (File S4). ENO1 was chosen for additional analysis based on the MS results since it is a crucial enzyme in glycolysis. RIP assays were carried out in TU177 and AMC-HN-8 using the ENO1 antibody and the lgG antibody as a negative control. NMRAL2P was enriched in the ENO1 antibody group and was around 20 times greater than in the lgG group (Fig. 4B). To investigate the precise site of NMRAL2P binding to protein ENO1, a truncated plasmid tagged ENO1 with a flag tag was used to divide ENO1 into three segments: ENO1-A (1-97aa), ENO1-B (97-237aa), and ENO1-C. (237-434aa). ENO1 truncation efficiency was verified using western blot (Fig. S1). RIP experiment results showed that NMRAL2P is mainly enriched in the ENO1-C group (Fig. 4C). NMRAL2P-oe and NMRAL2P-ASO and their corresponding vectors were then transferred into TU177 and AMC-HN-8 cells, respectively, and the difference in ENO1 protein content was analyzed. Upregulated NMRAL2P raised ENO1 protein content compared with the vector, and downregulated NMRAL2P lowered ENO1 protein content compared with NC (Figs. 4D, 4E). To verify the results of previous studies that reported that RNA affects the protein content by affecting protein degradation, NMRAL2P-oe and NMRAL2P-ASO and their vectors were transferred into TU177 and AMC-HN-8, respectively, and cycloheximide was added to block protein synthesis. Proteins were collected at 0 h, 3 h, 6 h, 9 h, and 12 h and western blot was used to detect the protein level of ENO1. ENO1 was found to be more stable in the NMRAL2P overexpression group, whereas its half-life was reduced in the gene downregulation group (Figs. 4F–4G). The protein level of ENO1 in cells without cycloheximide did not decrease significantly after 12 h (Fig. S2). These results indicated that NMRAL2P can attach to ENO1 and aid in ENO1’s protein stability.

Figure 4 NMRAL2P combined with ENO1 to delay the degradation of ENO1.

(A) MS2-12x pulldown process. (B) The binding of ENO1 and NMRAL2P was verified by RIP assay (with lgG antibody as the negative control). (C) The full-length plasmid ENO1 with flag tag (ENO1-FL) and truncated plasmid (ENO1-A, ENO1-B, ENO1-C) were transfected into TU177 cells for the RIP assay, and QRT-PCR was used to detect the enrichment of NMRAL2P. (D–E) Western blot was used to verify the changes of ENO1 after NMRAL2P-oe and NMRAL2P-ASO transfection in TU177 and AMC-HN-8 cells. (F–G) Comparing the degradation of ENO1 after NMRAL2P-oe and NMRAL2P-ASO were transferred into TU177 and AMC-HN-8 cells (with Vector, ASO-NC as negative control) by Western blot assay. (RIP, RNA Binding Protein Immunoprecipitation; NC, NC-ASO, Negative control of NMRAL2P knockdown; Vector: pcDNA3.1; CHX, Cycloheximide).

NMRAL2P acts on ENO1 for tumor promotion

We transferred the ENO1 knockdown plasmid and used QRT-PCR and western blot to confirm knockdown effectiveness. In TU177 and AMC-HN-8 cells, the levels of ENO1 mRNA were reduced by more than 40% and the protein level of ENO1 in the shENO1 group was significantly lower than in the vector (pGenesil-1) group (Figs. 5A, 5B). After knocking down the levels of ENO1 mRNA, the migration, invasion, and proliferation abilities of tumor cells decreased (Fig. S3). Functional recovery assays were carried out to further clarify the role of NMRAL2P and ENO1 in tumor promotion. MTS, Transwell, and clone formation assays showed that NMRAL2P overexpression increased the viability, migration, invasion and proliferation of tumor cells, while ENO1 gene knockout partially counteracted this tumor promoting effect (Figs. 5C–5H). The overexpression of NMRAL2P led to an increase in lactic acid synthesis, and the downregulation of ENO1 led to a decrease in lactic acid production (Figs. 5I–5J), indicating that NMRAL2P acts on ENO1 to promote head and neck tumor cells.

Figure 5 NMRAL2P plays a role in promoting cancer through ENO1.

(A–B) Verification of shENO1 knockdown efficiency in TU177 and AMC-HN-8 cells by QRT-PCR and western blot. (C–H) MTS, Transwell, and colony formation demonstrated that knockdown of ENO1 partially attenuated the enhanced cell proliferation, migration, and invasion induced by overexpression of NMRAL2P in TU177 and AMC-HN-8 cells. (I–J) Knockdown of ENO1 partially attenuated the enhanced lactic acid yield induced by overexpression of NMRAL2P in TU177 and AMC-HN-8 cells. (NC: pcDNA3.1 + pGenesil-1, nc + NMRAL2P-oe: pGenesil-1 + NMRAL2P-oe).

NMRAL2P promotes GPX2 transcription through Nrf2

Patients with head and neck tumors in the TCGA data set were divided into high- and low- expression groups according to the median expression of NMRAL2P, and then a “limma” differential analysis was performed between the two subgroups. A total of 144 differential mRNA were obtained, including 116 upregulated mRNA, and 28 downregulated mRNA. GPX2 mRNA was strongly expressed in the high NMRAL2P expression group and the difference multiple was the largest in this group (logFC =3.17; p < 0.001; Fig. 6A). The expression of GPX2 increased when NMRAL2P-oe was transferred into TU177 and AMC-HN-8 cells and the expression of GPX2 decreased when NMRAL2P was knocked down (Figs. 6B, 6C). Previous studies have shown that Nrf2 is the transcription factor of GPX2 (Baird & Yamamoto, 2020; Li, Jiang & Wu, 2020). QRT-PCR was used to verify the effectiveness of Nrf2’s overexpression (Fig. 6D) and to confirm shNrf2 construction. The expression of Nrf2 was over 40% lower compared to the vector (Fig. 6E). The expression of GPX2 increased or decreased, respectively, when Nrf2-oe and shNrf2 were transferred into TU177 and AMC-HN-8 cells (Figs. 6F, 6G), indicating that Nrf2 acts upstream of NMRAL2P. The relative expression of NMRAL2P and Nrf2 was also proportional to the relative expression of GPX2 in the TCGA data set (Figs. 6H, 6I). RIP experiments were carried out in TU177 and AMC-HN-8 with Nrf2 antibody to further investigate the role of NMRAL2P in Nrf2’s ability to boost the transcription of GPX2. The results showed that NMRAL2P was enriched in the Nrf2 antibody group (Fig. 6J), indicating NMRAL2P and Nrf2 work in combination. NMRAL2P-oe and pcDNA3.1 plasmids were transferred into TU177 cells, and ChIP assay was carried out with a Nrf2 antibody. The GPX2 promoter region (antioxidant response element; ARE; -GTGACTCAGTG-) showed a statistically significant increase in the NMRAL2P overexpression group compared with the vector group (Fig. 6K). These findings indicated that NMRAL2P can combine with Nrf2 to deliver Nrf2 into the GPX2 promoter region, enhancing the transcription of GPX2.

Figure 6 NMRAL2P promotes the transcription of GPX2 by binding to Nrf2.

(A) Differential analysis between the high- and low- expression groups of NMRAL2P in the TCGA dataset. (B) NMRAL2P was overexpressed in TU177 and AMC-HN-8 cells, and GPX2 changes were verified by QRT-PCR. (C) NMRAL2P-ASO was transfected into TU177 and AMC-HN-8 cells and the change of GPX2 was verified by QRT-PCR. (D–E) Verification of overexpression and knockdown efficiency of Nrf2-oe and shNrf2 in TU177 and AMC-HN-8 cells by QRT-PCR, respectively. (F–G) To verify the changes of GPX2 after overexpression or knockdown of Nrf2 in TU177 and AMC-HN-8 cells by QRT-PCR, respectively. (H) The relative expression of NMRAL2P is proportional to the relative expression of GPX2. (I) The relative expression of Nrf2 is proportional to the relative expression of GPX2. (J) The relationship between NMRAL2P and Nrf2, verified by RIP assay. (K) NMRAL2P overexpression increases the amount of the Nrf2 binding GPX2 promoter region (ARE, Antioxidant response element, (-GTGACTCAGTG-)). (L) The pathway of enrichment of differentially expressed genes in NMRAL2P high and low expression groups. (KEGG, Kyoto Encyclopedia of Genes and Genomes; GO, Gene Ontology).

GPX2 knockdown can reverse the tumor-promoting effect of NMRAL2P

Functional recovery assays were performed in order to confirm the contribution of NMRAL2P to the operation of GPX2. A GPX2 knockdown plasmid was constructed with greater than 30% efficiency (Figs. 7A, 7B). After knocking down the levels of GPX2 mRNA, the migration, invasion, and proliferation abilities of tumor cells decreased (Fig. S3). NMRAL2P-oe was transferred into TU177 and AMC-HN-8 cells and the results showed that the overexpression of NMRAL2P enhanced the proliferation, migration, and invasion of tumor cells. However, when shGPX2 was transferred into TU177 and AMC-HN-8 cells, this tumor-promoting effect was diminished (Figs. 7C–7H). ROS and MAD decreased concurrently with the overexpression of NMRAL2P, and ROS and MAD essentially returned to normal after GPX2 was downregulated (Figs. 7I, 7J). SOD activity showed the opposite trend. These results demonstrate that GPX2 knockdown can reverse the tumor-promoting effect of NMRAL2P. To further verify that the effect of NMRAL2P on tumors is the result of oxidative stress, an oxidative stress inhibitor, Tempol, was used to reduce the effect of oxidative stress. The results showed that the decrease in the migration, invasion, and proliferation of head and neck tumor cells caused by GPX2 knockdown was reversed by Tempol (1 mM; Fig. S4).

Figure 7 NMRAL2P plays a role in promoting cancer through ENO1.

(A–B) Verification of shGPX2 knockdown efficiency in TU177 and AMC-HN-8 cells by QRT-PCR. (C–H) MTS, Transwell, and colony formation demonstrated that knockdown of GPX2 partially attenuated the enhanced cell proliferation, migration, and invasion induced by overexpression of NMRAL2P in TU177 and AMC-HN-8 cells. (I–J) In TU177 and AMC-HN-8 cells, the knockout part of GPX2 gene enhanced the reduction of oxidative stress caused by NMRAL2P overexpression. (NC: pcDNA3.1 + pGenesil-1, nc + NMRAL2P-oe: pGenesil-1 + NMRAL2P-oe).

NMRAL2P overexpression enriches the ROS-related pathways

DE mRNA were identified by dividing NMRAL2P into a high and a low expression group based on the median. KEGG and GO enrichment analyses were then used to examine the identified DE mRNA. The KEGG enrichment analysis identified the processes that were enriched in the DE mRNA, including glutathione metabolism, pentose and glucuronate interconversions, and chemical carcinogenesis-reactive oxygen species. The GO enrichment analysis identified the following pathways: response to a toxic chemical, response to a xenobiotic stimulus, and response to a metabolic activity (Fig. 6L). The mechanism diagram is shown in Fig. 8.

Figure 8 Downstream Mechanism Diagram of NMRAL2P.

Discussion

This study identified 16 DE lncRNA related to oxidative stress through differential and correlation analyses of 546 head and neck cancers from TCGA. The differential expression and effect of these 16 DE lncRNA on prognosis was verified using samples from the library of the Second Hospital of Hebei Medical University. Only NMRAL2P was substantially expressed in cancer tissues and predicted a poor prognosis. NMRAL2P is primarily found in the nucleus. Functional assays in vitro show that NMRAL2P promotes the migration and proliferation of head and neck tumor cells, promotes lactic acid and SOD, and reduces ROS and MAD production. Western blot assays and further mass spectrometry analysis confirmed that NMRAL2P bound to protein ENO1 and prevented ENO1 from degrading. In the group with high NMRAL2P expression, the expression level of GPX2 was higher, and the difference multiple was the highest. Further assays confirmed that NMRAL2P promotes the transcription of GPX2 by binding to Nrf2. These results indicate that NMRAL2P can: promote the occurrence of glycolysis by binding to protein ENO1, provide tumor cells with energy, and subsequently promote tumor cell migration, invasion, and immune escape. By acting on GPX2 and scavenging excessive oxygen free radicals, NMRAL2P can also shield tumor cells from oxidative injury.

ENO1 (enolase), also referred to as 2-phosphate-D-glycerol hydrolase, is an essential enzyme in the glycolysis process as it catalyzes the transformation of 2-phosphate-glyceric acid into phosphoenolpyruvate. ENO1 has been shown to be highly expressed in tissue samples from a variety of tumor types (Huang et al., 2022b; Gou et al., 2021; Hou et al., 2021). High ENO1 expression predicts poor prognosis in patients with liver cancer and lung cancer and also indicates more severe tumor type and later stage tumors (Zhang, Lu & Yang, 2020; Zhang, Wang & Dong, 2018). Breast cancer lymphatic metastasis and ENO1 are also linked (Alagundagi et al., 2023). ENO1 can increase the process of glycolysis, help prevent immune cell destruction and the role of growth inhibitors in tumors (Hsiao et al., 2013), promote the growth of tumor blood vessels, provide more nutrients to cells, promote tumor growth and migration (Huang et al., 2022a), promote malignant proliferation, and act on tumor-related pathways (Hua et al., 2021). The increased expression of ENO1 in all malignancies signifies the heightened potential of migration and invasion, which relies on the type of tumor and the subcellular localization of ENO1 (Ejeskär et al., 2005). In this study, ENO1 was abundantly expressed in cancer tissues and was correlated with poor prognosis. The cancer-promoting effect of NMRAL2P can be weakened by ENO1 knockdown. These results indicate NMRAL2P plays a cancer promoting role, promoting glycolysis by binding to the ENO1 protein.

Glycolysis activation also produces a significant amount of ROS, resulting in the disruption of the oxidant/reductant balance, leading to DNA and protein damage, and ultimately cell death (Lan et al., 2020). By promoting the production of reducing agents, tumor cells can lessen the harm caused by oxidants such as ROS. It is well known that GPX2 is a significant reductant in tumor tissues and is substantially expressed in various malignancies (Esworthy, Doroshow & Chu, 2022; Yang et al., 2022). GPX2 can promote the migration and invasion of prostate, pancreatic, and ovarian cancer cells through the Wnt/β-catenin/EMT pathway (Li, Dai & Niu, 2020; Wang et al., 2019; Zhang et al., 2018). This study also confirmed that NMRAL2P can promote the transcription of GPX2 and neutralize excessive ROS. GPX2, NQO1, and HO-1 are downstream of Nrf2 (Rojodela Vega, Chapman & Zhang, 2018).

The effect of Nrf2 on several genes in tumor cells is mostly related to the removal of oxidants and the restoration of the oxidation/reduction balance in these cells. Nrf2 binds to Keap1 when oxidative stress is not present. When oxidative stress is triggered, Nrf2 is released from Keap1 and transferred into the nucleus, where it promotes the transcription of genes related to oxidative stress, neutralizing ROS by generating a large number of genes that reduce ROS, thus keeping the body’s oxidation and reduction levels balanced (Kansanen et al., 2013). In addition to directly promoting the transcription of related mRNA, Nrf2 can also promote the transcription of long noncoding RNAs such LUCAT1, ODRUL, RoR, and TUG1 (Luzón-Toro et al., 2019; Gao et al., 2017; Zhang et al., 2014; Yang et al., 2020). Prior research has also confirmed that NMRAL2P is downstream of Nrf2 and can promote the transcription of NQO1 and neutralize excess oxidants. Nrf2 is not only the main gene for oxidative stress (Johnson et al., 2017; Bhattacharjee, Li & Dashwood, 2020), but it can also act on p53 and prevent apoptosis (Puca et al., 2009). In addition, it can act on the wtp53 factor, have a carcinogenic role, and promote the immunological escape of tumor cells (Aubrey et al., 2018). This study confirmed that NMRAL2P can boost Nrf2’s role in promoting the transcription of GPX2.

Conclusion

The purpose of this study was to analyze the role of lncRNA related with oxidative stress in head and neck tumors. High expression of NMRAL2P in head and neck malignancies predicted poor prognosis. Functional experiments showed that NMRAL2P can promote the production of lactic acid and SOD, reduce the production of ROS and MDA, and increase the invasion and migration abilities of head and neck tumor cells. The cytoplasmic portion of NMRAL2P can bind to the ENO1 protein, postpone ENO1’s breakdown, promote glycolysis, and provide extra energy to tumor cells. The gene can also boost Nrf2’s role in promoting the transcription of GPX2, neutralize excessive oxidation production, and prevent tumor cells from being destroyed by oxidants.

Supplemental Information

Supplemental Information 1 The effects of NMRAL2P overexpression and knockdown on the half-life of ENO1 were analyzed by Western blot

NMRAL2P overexpression plasmids (NMRAL2P-OE) or NMRAL2P knockdown (NMRAL2P-ASO) and their corresponding negative control groups (Vector or NC) were transferred into TU177 and AMC-HN-8 cells. After adding cycloheximide,proteins were collected at 0 h, 3 h, 6 h, 9 h and 12 h, respectively. Western blotting was used to detect the changes of ENO1 protein level to verify the effect of overexpression of NMRAL2P or knocking down NMRAL2P on ENO1 degradation. The protein blot images of ENO1.

Click here for additional data file.

Supplemental Information 2 The effects of NMRAL2P overexpression and knockdown on the protein level of ENO1 were analyzed by Western blot

NMRAL2P overexpression plasmids (NMRAL2P-OE) or NMRAL2P knockdown (NMRAL2P-ASO) and their corresponding negative control groups (Vector or NC) were transferred into TU177 and AMC-HN-8 cells. 48 h later, the expression of ENO1 was detected. The image of protein ENO1 blot after NMRAL2P-oe or NMRAL2P-ASO were transferred into TU177 and AMC-HN-8 cells.

Click here for additional data file.

Supplemental Information 3 Domain inquiry Fig. 4C

Click here for additional data file.

Supplemental Information 4 Code

Click here for additional data file.

Supplemental Information 5 The sequence of primers and shRNA

Click here for additional data file.

Supplemental Information 6 1399 mRNAs associated with oxidative stress

Click here for additional data file.

Supplemental Information 7 Verifying ENO1 truncation efficiency by western blot

Click here for additional data file.

Supplemental Information 8 Verification of the effectiveness of cycloheximide

(A–B) After TU177 and AMC-HN-8 cells transfected into NMRAL2P-oe,NMRAL2P-ASO and the corresponding negative control groups Vector (pcDNA3.1), NC (NC-ASO), the changes of ENO1 protein were compared by western blot after 12 h with and without cycloheximide.(CHX: cycloheximide)

Click here for additional data file.

Supplemental Information 9 ENO1 and GPX2 knock down function assay

(A–B) The changes of cell viability of TU177 and AMC-HN-8 after transferred into shENO1 and shGPX2 were verified by MTS assay. (C–D) the changes of cell migration and invasion ability of TU177 and AMC-HN-8 after transferred into shENO1 and shGPX2 were verified by transwell assay. (E–F) the changes of proliferation activity of TU177 and AMC-HN-8 after transferred into shENO1 and shGPX2 were verified by clone formation assay. (G) The changes of lactate levels in TU177 and AMC-HN-8 cells after transferred into shENO1 were verified by lactic acid kit. (H) The changes of ROS yield, SOD activity and MAD content in TU177 and AMC-HN-8 cells after transferred into shGPX2 were verified. (ROS, Reactive Oxygen Species; SOD, Superoxide Dismutase; MAD, Malondialdehyde)

Click here for additional data file.

Supplemental Information 10 Functional recovery assays

(A–F) MTS, Transwell, and colony formation experiments showed that Tempol partially enhanced the reduced migration, invasion, and proliferation ability of TU177 and AMC-HN-8 cells induced by GPX2 knockdown. (G–H) In TU177 and AMC-HN-8 cells, Tempol partially attenuated the enhancement of oxidative stress induced by shGPX2 knockout. (ROS, Reactive Oxygen Species; SOD, Superoxide Dismutase; MAD, Malondialdehyde; Tempol, A superoxide dismutase analogue that effectively neutralizes reactive oxygen species)

Click here for additional data file.

Supplemental Information 11 The verification results of NMRAL2P overexpression plasmid

Click here for additional data file.

Supplemental Information 12 The sequence of NMRAL2P-ASO, shENO1 and shGPX2 and the sequencing results of shENO1 and shGPX2

Click here for additional data file.

Supplemental Information 13 Verification of truncated plasmid by ENO1

Click here for additional data file.

Supplemental Information 14 Original mass spectrometry results

Click here for additional data file.

Supplemental Information 15 Raw data of supplementary assays

Click here for additional data file.

The authors are grateful to Prof. Wang Baoshan for guidance on the design of this study.

Additional Information and Declarations

Competing Interests

Author Contributions

Human Ethics

Data Availability

The authors declare there are no competing interests.

Qian Nie performed the experiments, analyzed the data, authored or reviewed drafts of the article, and approved the final draft.

Huan Cao analyzed the data, authored or reviewed drafts of the article, and approved the final draft.

JianWang Yang performed the experiments, prepared figures and/or tables, and approved the final draft.

Tao Liu performed the experiments, prepared figures and/or tables, and approved the final draft.

BaoShan Wang conceived and designed the experiments, prepared figures and/or tables, authored or reviewed drafts of the article, and approved the final draft.

The following information was supplied relating to ethical approvals (i.e., approving body and any reference numbers):

The informed consent and the consent procedure gained approval from the Research Ethics Committee of the second hospital of Hebei Medical University(Ethical Application Ref:2021-R012-01)

The following information was supplied regarding data availability:

The raw data, including Western blots, are available at Figshare:

Nie, Qian (2023). Raw data , western blot and code.zip. figshare. Dataset. https://doi.org/10.6084/m9.figshare.22678120.v1

The code and additional raw data are available in the Supplemental Files.

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
