# Peer review of "Long non-coding RNA NMRAL2P promotes glycolysis and reduces ROS in head and neck tumors by interacting with the ENO1 protein and promoting GPX2 transcription"

_PeerJ, doi:10.7717/peerj.16140_

## Round 0.1 · original submission · Major Revisions

We have received detailed remarks from three reviewers. All the reviewers suggested revision. Please pay attention to the figures' quality. You are welcome to resubmit the revised manuscript.

Reviewer 1 ·

Basic reporting

In this work, Nie et al proposed that NMRAL2P enhances cancer phenotype by promoting transcription of GPX2 via binding to Nrf2 and inhibiting degradation of ENO1. In addition, they also proposed NMRAL2P enhances cancer phenotype by increasing the production of lactic acid and reducing oxidative stress. I would suggest the authors improve their manuscript following the comments below:
1. This manuscript needs extensive improvement in grammar and scientific writing. It is not written in a professional way. Many terms are not used in the right places. The information that the authors want to tell is not delivered adequately due to language issues.
2. Better proofreading will be needed for this manuscript. Many typos in the text and mislabeling in the figures are found.
3. Statistical method issues. For example, there are two variables (time and treatment) in the plots of proliferation assays. Neither t test nor the chi-square test that the authors mentioned in the methods session would be adequate to analyze the data.
4. The Method session in this manuscript is extremely succinct and very hard to follow.
5. The authors want to show knockdown NMRAL2P increases oxidative stress by showing increased ROS. However, I would suggest the authors use oxidative stress markers to determine if oxidative stress is really affected. In fact, the authors did not show if the changes in cell phenotype are mediated by oxidative stress. I would suggest using either oxidative stress inducer or inhibitor to antagonize either NMRAL2P overexpression or knockdown action.
6. It will be important to show if modifying the level of ENO1 cause any changes in cell phenotype. This also applies to GPX2. Meanwhile, it will be interesting to know if changing the level of either ENO1 or GPX2 changes oxidative stress in cells.
7. I did not see any data showing the interaction between Nfr2 and GPX2 promoter. The conclusion that NMRAL2P binds to Nrf2 and delivers Nrf2 to GPX2 promoter region is not well supported.
8. A control will be needed to show if cycloheximid is actively blocking protein synthesis in Figure 4. And according to the data in Figure 4F and 4G, their NC reduces protein degradation compared to the vector control. This means the controls have effects on target protein degradation. How would the authors explain that?

Experimental design

Some improvements will be needed. Please see the details in the basic reporting.

Validity of the findings

Please see details in the basic reporting.

·

Basic reporting

1. The English language should be improved to ensure that an international audience can clearly understand your text. Some examples where the language could be improved include lines 113-114, 263, 270-271, 278-279, 288, 300, 350 – the current phrasing makes comprehension difficult. I suggest you have a colleague who is proficient in English and familiar with the subject matter review your manuscript, or contact a professional editing service.

2. Please pay more attention to some minor mistakes, such as line 302, which should be "(Fig.7A-B).". Line 346, should be "weaken". Line 166, it should be 1:10000 instead of 1 10000. Line 133, why do you use 2103 cells? Should it be 2*10^3 (2000) cells?

3. Line 282, please add the reference for "The transcription factor of GPX2 is Nrf2, according to earlier research."

4. Line 289 and Figure 6I, please use the same name for Nrf2 instead of NFE2L2 in the figure to make it easier to understand.

5. Why do you use "Nrf2" instead of "NRF2" in the manuscript?

6. Please unify the presentation of the figures, such as line 196. Please use a consistent format, either "Fig.1A, Fig.1B" or "Fig1A, Fig1B".

7. In the figure descriptions, it would be beneficial to maintain consistency by using either a semicolon ";" or a period "." at the end of each sentence. Additionally, for clarity, please use a consistent format for the subfigure labels, either "(A)" or "(A):".

8. Figure 3, please explain the full name of "ASO". You can include it in the main body.

9. Figure 4C, please detect the level of ENO1 truncations with experiments such as Western blot.

10. Figure 6K, what is "ARE"?

11. Figure 6L, please enlarge the font. It is too small to read the text.

Experimental design

The experiment is well-designed to support their conclusion.

Validity of the findings

1. In Figure 4F and 4G, it is noticeable that the expression of ENO1 appears to be different in the two control groups (vector and NC). Please provide an explanation for this observed difference in expression levels.

2. For Figures 5A and 5B, I would suggest considering the inclusion of additional experimental techniques, such as Western blot analysis, to detect the protein levels of ENO1. Supplementing the current methodology with protein-level analysis would provide a more comprehensive understanding of ENO1 expression and strengthen the reliability and significance of the results.

Additional comments

The authors have effectively employed a combination of bioinformatics analysis, cell experiments, and statistical analysis in patients to investigate the role of NMRAL2P in head and neck tumor pathogenesis and prognosis. The integration of these diverse approaches lends strength and comprehensiveness to the study's findings. Overall, this research demonstrates considerable potential for publication, with the recommendation for some optimization to further enhance its quality and impact.

Reviewer 3 ·

Basic reporting

1. I noticed the following grammatical or spelling errors:
a) Line 39: showed
b) Line 75: ‘differential’ instead of ‘difference’. Please check for this error all throughout the manuscript (for eg. Fig 6A legend).
2. Lines 49 to 52 need a citation.
3. Line 67: introduce the abbreviation ‘lncRNA’ here.
4. Line 76: You did not introduce the abbreviation OS-DElncRNAs.
5. Introduction should mention which cell lines are being used and why.
6. All of the results section needs to be corrected for standard nomenclature: when referring to gene or RNA please italicize and mRNAs are in small letters (eg: Eno1) and when referring to protein use capital letters and no italics (eg. ENO1).

Experimental design

1. Line 85: unclear if the matching samples means normal tissue?
2. Lines 117-118: unclear what the author means by three intricate holes and how that reduces error.
3. Lines 12-125: Explanation of Matrigel coating of the transwells is not clear and very confusing.
4. Line 125: 10% fetal bovine serum containing medium
5. Lines 126-129: Please rewrite for better clarity and use full-stop instead of comma. Not sure what the authors mean by 20 cells on line 129.
6. Line3 130-131: Unclear if ‘number of migrated and invaded cells’ were counted and how.
7. Line 143: Unclear what is micromethod? Do you mean a multi-plate reader?
8. Line 146: You did not introduce the abbreviation. Is it the ROS kit reagent?
9. Lines 241-243: The authors need to provide more details about the mass spectrometry technique and analyses methods used.

Validity of the findings

1. Figure 3 -please specify what ‘NC’ stands for in the figure legend.
2. Figure 4: Providing a quantification of the western data with statistical analyses will provide stronger proof.
3. Figure 4B – is there a difference in the endogenous levels of ENO1 between the two different cell lines?
4. Unclear if figure 5A is referring to RNA or protein expression levels. Also mention in figure legend.
5. Line 190: I think the title should be rephrased to ‘Identifying lncRNAs involved in regulating oxidative stress’.
6. Fig 6A legend should specify the dataset used instead of a description of results such as GPX2 sentence.
7. Fig 6K – please specify what ‘ARE’ stands for.

Additional comments

Overall, I think this is a good in vitro mechanistic study for assessing the oncogenic role of a lncRNA. With the various functional assays and rescue experiments, the authors provide enough proof to substantially validate their findings.

---

## Round 0.2 · Minor Revisions

The manuscript got remarks on the language presentation from both reviewers. It needs at least minor revision.

I see that overall logic of presentation should be updated - use short direct phrases, and avoid redundant abbreviations, especially in the Abstract.
Here are additional editorial comments:
See in the Abstract:
Lines 21-23 in the Reviewing PDF file:
Background:More reactive oxygen species (ROS), which promote the development of oxidative stress, will be produced as a result of metabolic reprogramming, a key hallmark of tumor.
- Split this phrase into 2 sentences. Not start phrase from ‘more’ word.
- Add phrase about importance of lncRNA as the object of the study.
Then (line 26 ) give abbreviation TCGA in full, describe the data set used.
Line 29: “samples from our sample library…” – rephrase, not use ‘sample’ twice, describe what mean ‘our library’.
Line 34 and below: “(QRT-PCR)… (ChIP) …” – remove such abbreviations, if not used more in the Abstract.
Use word ‘gene’ for NMRAL2P when discussing gene.
Line 39: “As indicated by the result of the In vitro functional experiments…” -
Not use capital letter for ‘In’, use Italic font for ‘in vitro’.
Remove wording “As indicated by the result” in the Results paragraph of the Abstract.
Line 53: “tumor selects glycolysis as the primary way to obta in energy[1-2]. .” – use proper grammar.
Not write ‘tumor selects’ – tumor is just a process, how it can select? Rephrase like “Glycolysis is the primary way to obtain energy in tumor…”
Line 60: “cancer tissue samples in previous studies [5-7],…” - add detail about ‘previous’ studies.
What cancer types were used? Not head and neck?
Then avoid redundant capital letters (like in word ‘Differentially’)
Write details for companies – city, country, company name – see “…Company (Shanghai, China)” and “(Vector)”
Check again front formatting – it varies in the PDF now. Should be uniform..
Try to minimize new abbreviations like “OS-DElncRNAs” through the text.
Use standard phrase in Acknowledgment like “The authors are grateful to Prof…”
Not “Thank you, Prof…”
Line 472: “The datasets analysed during the current study are available in the GDC repository[TCGA-473 HNSC] [https://portal.gdc.cancer.gov/].” –
- Please indicated the dataset number

Give proper references to the databases used, see
“retrieved from Genecards…” – cite GeneCards.org (online link, and literature reference too), describe search request to this database..

Figures’ quality (resolution) should be checked.
Please add short description what one can see in each Figure.
Currently, the references describing figure panels are too short.
For example “Differentially expressed lncRNAs(DElncRNAs):451 (up-regulation lncRNAs:252; down-regulation lncRNAs:199) were obtained(Fig.1A, B).”
- add details – that it is volcano plot there, describe Panel A and Panel B separately,
What is about DEmRNA and DElncRNA in the figure panels?
Write, at least, ‘DE mRNA’ with space. What is about comparison of differentially expressed mRNA and lncRNA numbers?

Overall, revision is necessary.

**Language Note:** The Academic Editor has identified that the English language must be improved. PeerJ can provide language editing services - please contact us at [email protected] for pricing (be sure to provide your manuscript number and title). Alternatively, you should make your own arrangements to improve the language quality and provide details in your response letter. – PeerJ Staff

Reviewer 1 ·

Basic reporting

The authors have improved their manuscript in methodology and scientific level. However, the language remains to be an issue. I would strongly suggest the authors seek help from a scientific paper editing service before resubmission.

Experimental design

No comment

Validity of the findings

No comment

·

Basic reporting

All the concerns of reviewer 2 have been answered. I suggest the authors spend more time on languages and details next time.

Experimental design

All the concerns of reviewer 2 have been answered. I suggest the authors spend more time on languages and details next time.

Validity of the findings

All the concerns of reviewer 2 have been answered. I suggest the authors spend more time on languages and details next time.

---

## Round 0.3 · accepted · Accept

Thanks for the revision and detailed answer. The reviewers and editors have no more critical remarks.

Reviewer 1 ·

Basic reporting

The authors improved their manuscript. I would recommend "Accept" this time.

Experimental design

No comment

Validity of the findings

No comment